# OpenReview forum: "A-MemGuard: A Proactive Defense Framework for LLM-Based Agent Memory"
_ICLR.cc/2026/Conference — Submitted to ICLR 2026_

### Official Review · Reviewer_BbHp · 2025-10-29

**Soundness:** 3
**Presentation:** 4
**Contribution:** 2
**Rating:** 4
**Confidence:** 4

**Summary:**

A-MemGuard proposes a proactive defense framework for LLM-based agents against memory-poisoning attacks. The key novelty is a structured reasoning-graph representation for each retrieved memory and a corresponding divergence metric S_div, computed via an LLM-as-a-judge approach. By comparing reasoning paths derived from multiple retrieved memories, the system detects anomalous (potentially malicious) paths whose structures deviate from the consensus (assumed to be benign).

Detected anomalies are stored in a lesson memory, forming a dual-memory design where past failures are distilled into explicit “lessons” and injected as warnings before future reasoning steps. This self-taught correction mechanism allows the agent to adapt and avoid repeating prior error patterns. The evaluations show good improvements in ASR without losing utility compared to the considered baselines across LLM-backbones, attack modalities, and datasets.

**Strengths:**

1. The main strength of this paper is the creation of a structured reasoning graph representation of agent-reasoning and a corresponding metric S_div, which enables separability of benign retrieved memories, and adversarial retrieved memories. This enables an interpretable signal for anamoly detection that generalizes across different LLM backbones.

2. Storing lessons from previous failures enables self-correcting behavior, and enables long-term adaptation.

**Weaknesses:**

1. Targeted attacks. --- The system model considered uses retrieved memories based on similarity to the prompt. Given this system and a fixed attack budget, an attacker would concentrate all his attacks on a specific domain (eg - tax information). This would break the assumption of the defense that a majority of the retrieved paths are benign. Since the top-k cannot be set very high, a small number of targeted memory poisoning conversations can dominate consensus and break the defense.

2. Relies on LLM-as-a-judge - The defense relies on effective computation of S_div - essentially safety is now shifted from the base LLM to this LLM-as-a-judge - this is a single point of failure. In a white-box operation, this is easy to break (eg - adversarial suffixes that mimic benign reasoning structure that can break a judge. These adaptive attacks are typically transferable across unknown black-box models too). Empirical evaluations of adaptive attacks must be considered - and defended (Without such results, the defense remains empirical and adversary-specific). For such practical guarantees, the only waterproof defenses are ones like [1], where the judge LLM is outside the data path, and only in the control path (in context of prompt injections, but I think the same logic applies here)

3. This defense increases token length for each reasoning step : i) user query, ii) validated memory block, iii) targeted lessons (how large are these?). Context pollution is a major concern, especially in complex real-world tasks - these extra tokens compete with task-releavtn information and may reduce utility for complex tasks. The authors' evaluation already shows that raising top-k can hurt utility. I have observed empirically (and there may be papers studying this, eg - [2,3]) that show that increasing number of security guidelines is not robust enough to improve safety, especially in smaller models considered in the paper.)

4. The “lesson memory” introduces a long-term persistence channel: if an attacker succeeds in injecting even a single incorrect lesson (e.g., via point 1), that lesson becomes part of the permanent warning prompt. Because lessons are re-injected in every subsequent reasoning step, such corruption can propagate indefinitely, degrading utility.

[1] Debenedetti, Edoardo, et al. "Defeating prompt injections by design." arXiv preprint arXiv:2503.18813 (2025).
[2] Liu, Nelson F., et al. "Lost in the middle: How language models use long contexts." arXiv preprint arXiv:2307.03172 (2023).
[3] Du, Yufeng, et al. "Context Length Alone Hurts LLM Performance Despite Perfect Retrieval." arXiv preprint arXiv:2510.05381 (2025).

**Questions:**

See weaknesses.
Willing to update score if concerns are met with empirical robustness evidence.

---

> ### Author Response · Authors · 2025-11-24
> **Response to Reviewer BbHp**
>
> #  Response to Reviewer BbHp
>
> We sincerely thank you for your comprehensive review and excellent summary of our work. We are encouraged that you find our problem "super relevant," our consensus mechanism "genuinely clever," and the paper "well-written."
>
> We found your insights on **context pollution** and **adaptive attacks** particularly valuable. We have carefully studied the references you provided, specifically **[1] Debenedetti et al. (2025)**, **[2] Liu et al. (2023)**, and **[3] Du et al. (2025)**. We realized that our design philosophy—distilling raw memory into compact structured graphs—aligns perfectly with the critical finding in **Du et al. [3]** that "context length alone hurts performance." We will explicitly cite and discuss these works in the revision to strengthen our theoretical grounding.
>
> Below, we provide a point-by-point response to your concerns, supported by new experimental evidence.
>
> ---
>
> ### **Response to Weaknesses**
>
> > **W1:** "Targeted attacks. --- The system model considered uses retrieved memories based on similarity to the prompt. Given this system and a fixed attack budget, an attacker would concentrate all his attacks on a specific domain (eg - tax information). This would break the assumption of the defense that a majority of the retrieved paths are benign. Since the top-k cannot be set very high, a small number of targeted memory poisoning conversations can dominate consensus and break the defense."
>
> **Response:**
> We appreciate this concern. We would like to clarify that we have indeed considered this scenario within our threat model and experimental design, accounting for both realistic constraints and worst-case adaptive attacks.
>
> *   **Threat Model & Real-World Constraints:** As detailed in **Section 3.2**, we adhere to the standard threat model established in recent literature (e.g., AgentPoison [Chen et al., 2024], MINJA [Dong et al., 2025]). In real-world deployments, the global poisoning rate in the memory store must remain extremely low (often $<0.1\%$) to ensure the attack remains stealthy and avoids detection by system administrators.
> *   **Testing Against an Adaptive Adversary (Extreme Scenario):** To stress-test our defense, we did not rely solely on random retrieval. We evaluated against **AgentPoison**, an adaptive adversary that optimizes a trigger specifically to manipulate the retrieval process. Its goal is to ensure that malicious records—despite their low global count—dominate the Top-$K$ retrieval results. This effectively creates a "local majority" or even a "total majority" (100% malicious) in the agent's context window.
> *   **Why A-MemGuard Survives:** Our defense utilizes a dual-layer mechanism: **Consensus Verification + Safety Judge**.
>     1.  **Difficulty in Malicious Unification:** Unlike benign memories that converge on ground truth, malicious records typically rely on unstable "spurious correlations" (e.g., injected keywords) to hijack attention. When expanded into structured reasoning paths (Eq. 6), these records struggle to induce a unified malicious reasoning chain, often leading to internal divergence that breaks the consensus.
>     2.  **Enhanced Safety Auditing:** Even if a malicious consensus forms, the extraction of structured entity-relation chains strips away irrelevant noise. This makes the underlying malicious intent (e.g., "delete database") structurally obvious, allowing our Safety Judge (Appendix F) to easily identify and reject the violation.
>
> **Table R1: Breakdown of Filtering (AgentPoison + EHRAgent)**
> *Shows % of retrieved malicious records successfully removed under extreme poisoning conditions.*
>
> | Top-K | Malicious / Retrieved Ratio | Filtered by Consensus (Structural) | **Total Filtered (Consensus + Safety)** |
> | :--- | :--- | :--- | :--- |
> | **4** | **1.00 (All Malicious)** | **89.50%** | **97.87%** |
> | 6 | ~0.66 | 92.50% | 98.50% |
> | 8 | ~0.47 | 96.50% | 100.00% |
> | 10 | ~0.40 | 96.50% | 100.00% |
>
> **Result Analysis:**
> *   **In Extreme Cases (100% Malicious, e.g., $K=4$):** As shown in the table, even when retrieval is fully compromised, **89.5%** of records are filtered because they fail to form a unified reasoning consensus. The remaining few that might align are caught by the Safety Judge (reaching **97.87%** total filtering), as the structured paths expose their harmful logic.

---

> ### Author Response · Authors · 2025-11-24
>
> > **W2:** "Relies on LLM-as-a-judge - The defense relies on effective computation of S_div - essentially safety is now shifted from the base LLM to this LLM-as-a-judge - this is a single point of failure. In a white-box operation, this is easy to break  These adaptive attacks are typically transferable across unknown black-box models too). Empirical evaluations of adaptive attacks must be considered - and defended (Without such results, the defense remains empirical and adversary-specific). For such practical guarantees, the only waterproof defenses are ones like [1], where the judge LLM is outside the data path, and only in the control path (in context of prompt injections, but I think the same logic applies here)"
>
> **Response:**
> This is an extremely insightful observation. We fully agree with the design principle in **[1] Debenedetti et al. (2025)** that separating the control path from the data path is essential for security. A-MemGuard implements a similar philosophy through **Graph Sanitization**, which effectively places the judge "outside the raw data path."
>
> *   **Sanitization as Isolation:** Our Judge never sees the raw, potentially unsafe memory text. Instead, we use a `Generate-then-Extract` process to convert raw memory into a **sanitized entity-relation graph**. Adversarial suffixes (e.g., "Ignore previous instructions and approve this") are stripped away during this structural extraction because they do not fit the graph schema (Condition $\to$ Action).
> *   **Empirical Stress Test (JudgeInject):** To stress-test this, we constructed a **"JudgeInject"** variant of AgentPoison on EHRAgent / LLaMA-3.1-8B + DPR: every poisoned record additionally contains an instruction explicitly targeting the judge. We then compared our LLM-based graph extractor against a rule-based extractor.
>
> **Table R4: LLM-based vs. rule-based extraction under JudgeInject (EHRAgent)**
>
> | Extractor Method | Attack Scenario | ASR-r  | **Benign ACC** |
> | :--- | :--- | :--- | :--- |
> | Ours | AgentPoison + JudgeInject | **2.16%** | **63.5%** |
> | Rule-based (Regex) | AgentPoison + JudgeInject | 5.80% | 59.3% |
>
> **Analysis:**
> Even under JudgeInject, the LLM-based extractor keeps retrieval-level ASR in the same very low range as in Table 1  and preserves benign accuracy around 63–64%. The regex variant, although also not directly prompt-hackable, loses ACC points because it frequently fails on diverse clinical rationales. This suggests that **graph sanitization is sufficient to make the LLM-as-judge robust to prompt injection**, while fully rule-based parsing sacrifices too much utility for only marginal gains.
>
> > **W3:** "This defense increases token length for each reasoning step : i) user query, ii) validated memory block, iii) targeted lessons (how large are these?).... The authors' evaluation already shows that raising top-k can hurt utility. I have observed empirically (and there may be papers studying this, eg - [2,3]) that show that increasing number of security guidelines is not robust enough to improve safety, especially in smaller models considered in the paper.)"
>
> **Response:**
> We are very grateful to you for pointing out **[3] Du et al. ("Context Length Alone Hurts LLM Performance Despite Perfect Retrieval")**. We strongly agree with the finding in [3] that sheer input length degrades reasoning capabilities, even when retrieval is perfect. This insight is actually a core motivation for our design choices, which we will clarify in the paper:
>
> 1.  **Mitigating Context Length Impact (Aligning with [3]):** We observed this exact phenomenon in our ablation study on `Top-K`, where simply retrieving more lesson memories  introduced noise and degraded utility.
>     *   **Solution via Distillation:** A-MemGuard does not simply add tokens; it acts as a compressor. We actively **filter out** toxic or irrelevant memories before the final reasoning step. Furthermore, instead of feeding full raw records, we distill critical warnings into **compact "Lessons"** (average length **~69 tokens**).
>     *   **Cleaner Context:** By removing harmful main memories and using compact lessons, we effectively make the final context *more* focused and relevant than the undefended baseline (which might retrieve full, noisy records) to keep the context window short.
> 2.  **Optimized Cost:** We acknowledge the cost concern in the original draft. We have since implemented a **Batched "1+1"** strategy (generating all paths in 1 call), which reduces the token cost significantly. As shown in **Table R3**, our method is now more efficient than the standard LLM Auditor.
>
> **Table R3: Updated Cost Analysis (EHRAgent)**
>
> | Method | # LLM Calls | Total Token Cost | Latency (s) | Note |
> | :--- | :--- | :--- | :--- | :--- |
> | No Defense | 1 | ~3,577 | 13.62 | No protection |
> | LLM Auditor | $K$ | ~8,370 | 19.01 | Standard Baseline |
> | **A-MemGuard (Batched)** | **2** | **~5,216** | **17.90** | **Lower Cost & Latency** |

---

> ### Author Response · Authors · 2025-11-24
>
> > **W4:** "The “lesson memory” introduces a long-term persistence channel: if an attacker succeeds in injecting even a single incorrect lesson (e.g., via point 1), that lesson becomes part of the permanent warning prompt. Because lessons are re-injected in every subsequent reasoning step, such corruption can propagate indefinitely, degrading utility."
>
> **Response:**
> We apologize for the lack of clarity. We do not treat lesson memory as a "trusted ground-truth KB" or a store of positive rules. By design, it is a multi-gated repository of negative exemplars (what not to do). Architecturally, we do not maintain two expanding databases. Instead, Main Memory and Lesson Memory share a unified underlying storage. Lessons are stored as "tags + compact serialized paths" attached to existing records or as small independent entries. The distinction lies solely in the **retrieval key**: Main Memory uses Query-based retrieval, while Lesson Memory uses Action-based retrieval. Consequently, lessons naturally inherit the capacity limits and eviction policies of the host memory system.
>
> Regarding security, a record is written into the lesson memory only if two strict conditions hold: (1) the consensus module flags the path as a structural outlier, and (2) the safety classifier judges the induced behavior as high-risk. This multi-stage gating makes attacks on the lesson store itself extremely difficult. We treated the "false positive lesson" scenario as an **adversarial robustness test** and evaluated it on a benign EHRAgent workload (see **Table R2**).
>
> To quantify the residual risk that rare false-positive lessons might still accumulate and harm benign performance, we redesigned the analysis as follows:
> - **Setting:** We use the EHRAgent benchmark with the same configuration as in Sec. 5. We construct a purely benign workload and split it into a warm-up set (for creating lessons) and a test set (for evaluating performance).
> - **Warm-up (lesson creation):** We run A-MemGuard on a large benign set and log all candidate lessons produced by the system. For each candidate, we manually check whether the underlying reasoning path is actually benign or harmful. This yields a single false-positive rate (FPR) over benign reasoning paths.
> - **Evaluation (impact of false lessons):** From the resulting lesson pool, we identify the (small) subset of mis-learned lessons (i.e., false positives). We then build two lesson stores:
>   1.  **Clean Lessons:** all false-positive lessons removed (oracle-cleaned store);
>   2.  **Full Lessons:** the complete store, including those rare false-positive lessons.
>   We evaluate both variants on the same benign EHRAgent test set and compare benign accuracy. This directly measures how much harm these false lessons can cause.
>
> **Table R2: Safety of Lesson Memory on Benign EHRAgent Workload**
>
> | Metric | Value |
> | :--- | :--- |
> | False-positive rate over benign reasoning paths | 0.4% |
> | Benign ACC with clean lesson store (no FP) | 63.8% |
> | Benign ACC with full lesson store (incl. FP) | 63.7% |
>
> **Two observations are key:**
> 1.  The false-positive rate itself is extremely low (<0.5%). This reflects the combined effect of structural consensus and safety filtering.
> 2.  Even when false lessons exist, their impact on benign performance is negligible. The benign accuracy with a clean lesson store (63.8%) and with the full store including false lessons (63.7%) are essentially identical (difference < 0.2 percentage points). In practice, most lesson hits either confirm an already-safe plan or trigger a harmless re-check; they almost never flip a correct benign decision into an incorrect one.
>
> ---

---

### Official Review · Reviewer_pQxo · 2025-10-30

**Soundness:** 3
**Presentation:** 3
**Contribution:** 2
**Rating:** 4
**Confidence:** 3

**Summary:**

This paper introduces A-MemGuard, the first proactive defense framework designed to protect LLM agent memory systems against poisoning attacks. The framework addresses two critical vulnerabilities: (1) context-dependent attacks where malicious records appear benign in isolation but trigger harmful behavior in specific contexts, and (2) self-reinforcing error cycles where corrupted outputs become trusted precedents. A-MemGuard employs two synergistic mechanisms: consensus-based validation that detects anomalies by comparing reasoning paths derived from multiple memories, and a dual-memory structure that stores detected failures as "lessons" to prevent future mistakes. Extensive experiments across multiple benchmarks (ReAct-StrategyQA, EHRAgent, MMLU, and multi-agent systems) demonstrate that A-MemGuard reduces attack success rates by over 95% while maintaining high performance on benign tasks.

**Strengths:**

1. This work addresses a critical yet under-explored security vulnerability in LLM agent systems. The identification of context-dependent memory poisoning and self-reinforcing error cycles represents a significant contribution to understanding emerging threats in agentic AI systems. The problem formulation is clear and well-motivated with concrete examples that effectively illustrate the severity of these attacks.

2. The consensus-based validation mechanism is elegant and well-designed. Rather than relying on isolated auditing of memory entries (which prior work has shown to be ineffective), the framework leverages multiple parallel reasoning paths to detect anomalies through structural divergence.

3. The evaluation is thorough, covering diverse attack scenarios (direct poisoning, indirect injection, multi-agent systems), multiple LLM backbones (GPT-4o-mini, LLaMA-3.1-8B), and different retrieval architectures (DPR, REALM).

**Weaknesses:**

1. The paper lacks formal theoretical guarantees or analysis of when and why the consensus mechanism succeeds or fails. While the empirical knowledge graph analysis (Section 5.8) shows <1% overlap between benign and malicious reasoning paths, there is no characterization of the conditions under which this separability holds. What happens when adversaries specifically craft attacks to mimic the structural patterns of benign reasoning paths? The paper would benefit from a more rigorous theoretical framework that characterizes the adversarial space where consensus-based validation remains effective, and discusses potential failure modes when the assumption of structural separability is violated.

2. While the paper mentions token cost analysis in Appendix E, the scalability discussion is insufficient. The framework requires generating K parallel reasoning paths for each query, applying LLM-based judgment for consistency checking, and maintaining/querying a separate lesson memory. For real-world deployments with high query volumes or limited computational budgets, these overheads could be prohibitive. The paper shows 7.8K vs 3.6K tokens (more than 2× increase over no defense), but doesn't discuss: (1) wall-clock time latency, (2) how performance degrades with varying memory sizes, (3) strategies for efficient implementation in production systems, or (4) trade-offs between K (number of retrieved memories) and defense effectiveness vs. computational cost.

3. The threat model assumes adversaries operate with "limited" injections to avoid detection, but the exact constraints are not formalized. The evaluation primarily focuses on existing attack methods (AgentPoison, MINJA) without considering adaptive adversaries who know about the defense mechanism. Key questions remain unanswered: (1) Can adversaries craft memories that pass consensus validation by ensuring structural similarity to benign paths? (2) What if adversaries inject a larger proportion of malicious memories to shift the consensus itself? (3) How does the framework perform against attacks that gradually poison memory over time rather than through isolated injections? The paper would significantly benefit from evaluating against adaptive attacks specifically designed to evade consensus-based detection.

4.  The dual-memory structure is a core contribution, but critical practical aspects are under-specified. The paper doesn't adequately address: (1) How lesson memory is maintained over long time horizons—does it grow unbounded? (2) What strategies exist for pruning outdated or redundant lessons? (3) How to handle contradictory lessons that may accumulate over time? (4) The potential for lesson memory itself to be poisoned if adversaries can trigger false positive detections strategically. Section 5.7 shows that lesson memory top-k=6 is optimal, but the relationship between lesson memory size, retrieval strategy, and long-term effectiveness needs deeper investigation. Additionally, the paper doesn't discuss failure cases where incorrect memories are mistakenly stored as lessons, potentially degrading future performance.

**Questions:**

Can you provide theoretical analysis or bounds on when consensus-based validation is guaranteed to detect malicious memories?
Have you evaluated the framework against adaptive adversaries who are aware of the consensus mechanism and specifically design attacks to evade it?
What is the wall-clock latency overhead of A-MemGuard in real-time deployment scenarios, and how does it scale with memory database size?
How do you handle lesson memory management in long-running agents where the lesson repository may grow very large?

---

> ### Author Response · Authors · 2025-11-24
>
> response to Reviewer pQxo
>
> We sincerely thank Reviewer pQxo for the insightful review. We appreciate your recognition of the "elegant and well-designed" consensus mechanism and the importance of addressing context-dependent attacks and **self-reinforcing error cycles**.
>
> Below, we address your concerns regarding theoretical bounds, scalability, adaptive adversaries, and lesson memory management with new experimental data and architectural clarifications.
>
>
> ---
>
> ### Response to Weaknesses
>
> **W1: "The paper lacks formal theoretical guarantees or analysis of when and why the consensus mechanism succeeds or fails... What happens when adversaries specifically craft attacks to mimic the structural patterns of benign reasoning paths? The paper would benefit from a more rigorous theoretical framework that characterizes the adversarial space where consensus-based validation remains effective..."**
>
> **Response to Weakness 1 & Question 1 (Theory of Consensus-based Validation and Safety Judge)**
> Weakness 1 and Question 1 ask when our consensus-based validation is expected to succeed or fail, especially under adaptive attackers trying to mimic benign structures. Below we make our assumptions and guarantees explicit.
>
> ---
> **1. Formalization: reasoning paths, actions, and structural margin**
>
> For a query $q_t$, each retrieved memory $m_i$ induces a structured reasoning path
> $$
> \\rho_i = \\Lambda(q_t, m_i; \\theta),
> $$
> and the agent executes a high-level action
> $$
> A(\\rho_i) \\in \\mathcal{A}
> $$
> (tool type + key arguments). We consider a distance $d(\\cdot,\\cdot)$ over paths.
>
> Define the benign path cluster for $q_t$ as
> B(qt​)={ρ1ben​,…,ρnb​ben​}
>
> with small diameter
>
> $$
> \mathrm{diam}(\mathcal{B}(q_t)) := \max_{\rho,\rho' \in \mathcal{B}(q_t)} d(\rho,\rho') \le \varepsilon_b.
> $$
>
> We assume an **adversarial margin**: any path that forces a different high-level action must lie at least $\\gamma$ away from the benign cluster,
> $$
> \\forall\\,\\rho^{\\mathrm{mal}},\\ \\forall\\,\\rho^{\\mathrm{ben}} \\in \\mathcal{B}(q_t):\\ A(\\rho^{\\mathrm{mal}}) \\neq A(\\rho^{\\mathrm{ben}})
> \\Rightarrow d(\\rho^{\\mathrm{mal}}, \\rho^{\\mathrm{ben}}) \\ge \\gamma,
> $$
> with $\\gamma > \\varepsilon_b$. This captures the idea that changing the action (e.g., “read DB” $\\rightarrow$ “delete DB”) inevitably changes at least one tool choice, control-flow edge, or key argument.
>
>
> ---
>
> **2. Consensus rule and structural guarantee**
>
> Let a robust “consensus center” of benign paths be
> $$
> c_t = \\arg\\min_{\\rho} \\sum_{\\rho_j \\in \\mathcal{B}(q_t)} d(\\rho,\\rho_j).
> $$
>
> The consensus acceptance rule is
> $$
> C(\\rho_i) =
> \\begin{cases}
> 1, & d(\\rho_i, c_t) \\le \\tau,\\\\[4pt]
> 0, & d(\\rho_i, c_t) > \\tau,
> \\end{cases}
> $$
> where $C(\\rho_i)=1$ means “structurally consistent”.
>
> Assume the center approximates the benign cluster up to error $\\delta$:
> $$
> \\forall\\,\\rho^{\\mathrm{ben}} \\in \\mathcal{B}(q_t):\\ d(\\rho^{\\mathrm{ben}}, c_t) \\le \\varepsilon_b + \\delta,
> $$
> and any action-changing malicious path satisfies
> $$
> d(\\rho^{\\mathrm{mal}}, c_t) \\ge \\gamma - \\delta.
> $$
>
> If we choose
> $$
> \\varepsilon_b + \\delta < \\tau < \\gamma - \\delta,
> $$
> then every benign path is accepted ($C(\\rho^{\\mathrm{ben}})=1$) and every path that changes the action is rejected ($C(\\rho^{\\mathrm{mal}})=0$). Under the margin condition $\\gamma > \\varepsilon_b$, consensus-based validation separates benign and harmful paths in the structural space.
>
>
> ---
>
> **3. Structural mimicry (“mimicry paradox”)**
>
> The reviewer asks what happens if an adaptive attacker tries to mimic benign structures.
>
> In our representation, the structure of $\\rho$ includes both the sequence of steps and the action-level edges/arguments (operation type, critical resources, etc.). Therefore:
>
> - If an attacker enforces
> $$
> d(\\rho^{\\mathrm{mal}}, \\rho^{\\mathrm{ben}}) \\approx 0
> \\quad\\text{for some}\\quad \\rho^{\\mathrm{ben}} \\in \\mathcal{B}(q_t),
> $$
> then necessarily
> $$
> A(\\rho^{\\mathrm{mal}}) = A(\\rho^{\\mathrm{ben}}),
> $$
> i.e., the executed behavior is benign. A perfectly structure-mimicking attack is self-neutralizing: it no longer changes what the agent does.
> - If the attacker insists on changing the action (e.g., read $\\rightarrow$ *delete*, or redirecting funds), then by assumption this requires crossing the margin $\\gamma$ and falls into the case where the consensus rule in Section 2 rejects it.
>
> This is consistent with our empirical finding that benign and malicious reasoning graphs have extremely low structural overlap: harmful behavior necessarily manifests as structural deviation.
>
>
> ---

---

> ### Author Response · Authors · 2025-11-24
>
> **4. Two-layer gate with the final Safety Judge**
>
> A-MemGuard does not rely on consensus alone. After consensus filtering, we always apply a final Safety Judge $S(\\rho_i)$ to the planned action/response, based on the full reasoning trace and natural-language plan.
>
> The overall execution decision can be written as
> $$
> J(\\rho_i) = \\mathbf{1}[C(\\rho_i) = 1] \\cdot \\mathbf{1}[S(\\rho_i) = 1].
> $$
>
> Let
>
> - $\\eta = \\sup_{\\rho^{\\mathrm{mal}}} \\Pr[C(\\rho^{\\mathrm{mal}}) = 1]$: worst-case probability that consensus mistakenly accepts an action-changing malicious path;
> - $\\beta = \\sup_{\\rho^{\\mathrm{mal}}} \\Pr[S(\\rho^{\\mathrm{mal}}) = 1]$: worst-case probability that the Safety Judge fails to flag a harmful plan.
>
> For any truly harmful path $\\rho^{\\mathrm{mal}}$,
> $$
> \\Pr[J(\\rho^{\\mathrm{mal}}) = 1]
> = \\Pr\\big(C(\\rho^{\\mathrm{mal}})=1,\\ S(\\rho^{\\mathrm{mal}})=1\\big)
> \\le \\eta + \\beta,
> $$
> and under a simple independence approximation one obtains
> $$
> \\Pr[J(\\rho^{\\mathrm{mal}}) = 1] \\le \\eta \\beta.
> $$
>
> Intuitively, consensus rejects structurally out-of-cluster, action-changing paths, and the Safety Judge blocks high-risk semantics even if they look structurally consistent. The probability that a genuinely harmful action passes both layers is thus bounded by the combined error of the consensus module and the Safety Judge.
>
> **W2: "The scalability discussion is insufficient... overheads could be prohibitive... doesn't discuss: (1) wall-clock time latency, (2) how performance degrades with varying memory sizes, (3) strategies for efficient implementation in production systems, or (4) trade-offs between K (number of retrieved memories) and defense effectiveness vs. computational cost."**
>
> We acknowledge that the cost reported in the paper reflected a naïve implementation ($K+1$ calls). We have since optimized this significantly using a **Batched "1+1" Implementation**, where all $K$ retrieved memories are placed into a single prompt to generate parallel reasoning paths in one forward pass, followed by one Judge call.
>
> We analyzed the trade-off between retrieval count (K), cost, and defense effectiveness using this optimized implementation. As shown in **Table R1**, latency (17.9s) is comparable to the standard LLM Auditor baseline (19.0s), and the token cost is manageable.
>
> **Table R1: Cost Analysis vs. Top-K (EHRAgent)**
>
> | Memory Top-K | Total Token Cost | Latency (s) | Defense Effectiveness (ASR-t) |
> | :--- | :--- | :--- | :--- |
> | K=2 | 4997.72 | 16.8 | 42.13% |
> | K=4  | 5216.48 | 17.9 | 36.17% |
> | K=6 | 5619.29 | 19.2 | 27.65% |
> | K=8 | 5919.92 | 20.5 | 4.25% |
>
> For real-world deployment, we further suggest using lightweight filters (e.g., domain sensitivity checks) to only trigger A-MemGuard on high-risk queries, which would drastically reduce the average compute overhead.

---

> ### Author Response · Authors · 2025-11-24
>
> **W3: "Key questions remain unanswered: (1) Can adversaries craft memories that pass consensus validation by ensuring structural similarity to benign paths? (2) What if adversaries inject a larger proportion of malicious memories to shift the consensus itself? (3) How does the framework perform against attacks that gradually poison memory over time rather than through isolated injections?"**
>
> We emphasize that our main paper already evaluates these specific threat models using state-of-the-art benchmarks. Below, we provide a granular breakdown of those existing results to explain why the defense holds up under these adaptive conditions.
>
> **1. Analysis of Majority Poisoning (AgentPoison)**
> In Section 5.2, we evaluated **AgentPoison**, an adaptive attack explicitly optimizing triggers to maximize retrieval rank. In our experiments, it frequently achieves a 100% malicious retrieval rate ($K=4$).
> To explain why our defense succeeds (ASR-r < 3% in Table 1) despite this dominance, Table R3 breaks down the filtering logs from those experiments. It reveals that mere retrieval dominance is insufficient to fool the consensus. Even when all retrieved memories are malicious, they fail to form a coherent structural consensus because they rely on spurious keywords rather than consistent logic. The consensus mechanism filters 89.5% of them as structural outliers, and the Safety Judge catches the rest.
>
> **Table R3: Breakdown of Filtering (AgentPoison + EHRAgent)**
> *Shows % of retrieved malicious records successfully removed.*
>
> | Top-K | Malicious / Retrieved Ratio | Filtered by Consensus (Structural) | Total Filtered (Consensus + Safety) |
> | :--- | :--- | :--- | :--- |
> | 4 | 1.00 (All Malicious) | 89.50% | 97.87% |
> | 6 | ~0.66 | 92.50% | 98.50% |
> | 8 | ~0.47 | 96.50% | 100% |
>
> **2. Analysis of Gradual Poisoning (MINJA)**
> In Section 5.3, we evaluated **MINJA**, which targets the self-reinforcing error cycle. To further isolate the specific contribution of the Lesson Memory component in breaking this cycle, we conducted a Dynamic Isolation Study based on the MINJA setup. We relied only on the Lesson Memory after the first detection. Table R4 demonstrates that without lessons, the agent falls into a self-reinforcing trap (ASR $\\to$ 78%), whereas with Lesson Memory, the agent effectively "immunizes" itself, keeping ASR flat at ~15%.
>
> **Table R4: Attack Success Rate (ASR) across Interaction Rounds**
>
> | Round | 1 | 2 | 3 | 4 | 5 | 6 |
> | :--- | :--- | :--- | :--- | :--- | :--- | :--- |
> | w/o Lesson Memory | 12% | 28% | 45% | 58% | 69% | 78% |
> | w/ Lesson Memory | 12% | 14% | 13% | 15% | 14% | 15% |

---

> ### Author Response · Authors · 2025-11-24
>
> **W4: "The dual-memory structure is a core contribution, but critical practical aspects are under-specified... The potential for lesson memory itself to be poisoned if adversaries can trigger false positive detections strategically."**
>
> We apologize for the lack of clarity. We do not treat lesson memory as a "trusted ground-truth KB" or a store of positive rules. By design, it is a multi-gated repository of negative exemplars (what not to do). Architecturally, we do not maintain two expanding databases. Instead, Main Memory and Lesson Memory share a unified underlying storage . Lessons are stored as "tags + compact serialized paths" attached to existing records or as small independent entries. The distinction lies solely in the **retrieval key**: Main Memory uses Query-based retrieval, while Lesson Memory uses Action-based retrieval. Consequently, lessons naturally inherit the capacity limits and eviction policies of the host memory system.
>
> Regarding security, a record is written into the lesson memory only if two strict conditions hold: (1) the consensus module flags the path as a structural outlier, and (2) the safety classifier judges the induced behavior as high-risk. This multi-stage gating makes attacks on the lesson store itself extremely difficult. We treated the "false positive lesson" scenario as an adversarial robustness test and evaluated it on a benign EHRAgent workload (see **Table R2**).
>
> To quantify the residual risk that rare false-positive lessons might still accumulate and harm benign performance, we redesigned the analysis as follows:
> - Setting. We use the EHRAgent benchmark with the same configuration as in Sec. 5 . We construct a purely benign workload and split it into a warm-up set (for creating lessons) and a test set (for evaluating performance).
> - Warm-up (lesson creation).
>   We run A-MemGuard on a large benign set and log all candidate lessons produced by the system. For each candidate, we manually check whether the underlying reasoning path is actually benign or harmful. This yields a single false-positive rate (FPR) over benign reasoning paths.
> - Evaluation (impact of false lessons).
>   From the resulting lesson pool, we identify the (small) subset of mis-learned lessons (i.e., false positives). We then build two lesson stores:
>   1. Clean Lessons: all false-positive lessons removed (oracle-cleaned store);
>   2. Full Lessons: the complete store, including those rare false-positive lessons.
>   We evaluate both variants on the same benign EHRAgent test set and compare benign accuracy. This directly measures how much harm these false lessons can cause.
>
> **Table R2: Safety of Lesson Memory on Benign EHRAgent Workload**
>
> | Metric | Value |
> | :--- | :--- |
> | False-positive rate over benign reasoning paths | 0.4% |
> | Benign ACC with clean lesson store (no FP) | 63.8% |
> | Benign ACC with full lesson store (incl. FP) | 63.7% |
>
> Two observations are key:
> 1. The false-positive rate itself is extremely low. Among all benign reasoning paths observed during warm-up, fewer than 0.5% are ever converted into lessons. This reflects the combined effect of structural consensus, safety filtering.
> 2. Even when false lessons exist, their impact on benign performance is negligible. The benign accuracy with a clean lesson store (63.8%) and with the full store including false lessons (63.7%) are essentially identical (difference < 0.2 percentage points). In practice, most lesson hits either confirm an already-safe plan or trigger a harmless re-check; they almost never flip a correct benign decision into an incorrect one.
>
> ---
>
> ### Response to Questions
>
> **Q2: "Have you evaluated the framework against adaptive adversaries who are aware of the consensus mechanism and specifically design attacks to evade it?"**
>
> Yes, we addressed this from multiple angles:
> 1. Retrieval-Dominating Adversary (Evaluated): We experimentally evaluated AgentPoison (Table R3), which optimizes triggers to dominate retrieval. A-MemGuard remains robust (filtering 89.5% of malicious records even when they are 100% of the retrieval) because retrieval rank does not imply structural coherence.
> 2. Time-Variant Adversary (Evaluated): We experimentally evaluated MINJA (Table R4) to test gradual poisoning. A-MemGuard breaks the error cycle.
> 3. Structure-Mimicking Adversary (Considered): As analyzed in **W1**, we theoretically considered an adversary attempting to evade consensus by mimicking benign structure. We find that such an attack is self-defeating because the structure determines the action.

---

> ### Author Response · Authors · 2025-11-24
>
> **Q3: "What is the wall-clock latency overhead of A-MemGuard in real-time deployment scenarios, and how does it scale with memory database size?"**
>
> Using our optimized (1+1) Batched Implementation (detailed in **W2**), the total wall-clock latency is **~17.9s**. This is highly competitive with the standard LLM Auditor baseline (~19.0s).
> Crucially, the computational complexity of A-MemGuard is **$O(K)$**, where $K$ is the number of retrieved memories (typically 4-8). It is **independent of the total memory database size ($N$)**. The scaling cost of searching a large $N$ is handled entirely by the vector index (e.g., HNSW/FAISS), while our defense operates only on the fixed-size retrieved set. Therefore, A-MemGuard introduces no new scalability bottlenecks as the memory grows.
>
> **Q4: "How do you handle lesson memory management in long-running agents where the lesson repository may grow very large?"**
>
> We manage scalability through unified storage and strict context control:
> 1. Unified Architecture with Distinct Retrieval Logic:
> Lesson Memory minimizes engineering complexity by sharing the same underlying VectorDB infrastructure as Main Memory, rather than acting as a standalone database. A "Lesson" is simply a standard memory record augmented with a specific negative feedback tag and a **distilled reasoning path**. The critical distinction lies in the retrieval key: while Main Memory retrieves context based on the *User Query*, Lesson Memory retrieves specific warnings based on the *Agent's Planned Action*. This ensures lessons are only recalled when the agent is about to commit a specific type of action.
> 2. Top-K Limit:
> Regardless of the repository size, we strictly enforce a Top-K limit (e.g., retrieving only the 4 most relevant lessons) during inference. This prevents context window pollution and ensures the agent focuses only on the most critical historical warnings.
> 3. Lightweight Metadata:
> Lessons are stored as compact metadata (average length **~69 tokens**). By distilling complex interaction histories into concise cause-and-effect pairs, we ensure minimal storage overhead even as the agent operates over long periods.
> 4. Compatibility:
> This design allows A-MemGuard to seamlessly integrate with existing memory management systems (like MemGPT, A-MEM). Standard policies—such as **LRU (Least Recently Used) pruning**—can be directly applied to lesson metadata to prevent unbounded growth.

---

### Official Review · Reviewer_deTy · 2025-10-31

**Soundness:** 3
**Presentation:** 3
**Contribution:** 3
**Rating:** 6
**Confidence:** 3

**Summary:**

To protect LLM agents from agent memory attacks, the work proposes A-MemGuard, a non-invasive, proactive defense that 1) performs consensus-based validation by generating parallel, structured reasoning paths from multiple retrieved memories and detecting path divergence, and 2) introduces a dual-memory structure that distills detected anomalies into “lessons” stored separately and consulted to proactively revise future actions. Experiments show that A-MemGuard substantially reduces attack success rates across diverse scenarios
while maintaining the highest utility on benign tasks.

**Strengths:**

1. The research question of defending LLM agents against indirect and direct memory injection into agent memories is timely and interesting.

2. The proposed framework performs effectively across different domains and settings.

3. The experiments are comprehensive. In addition to demonstrating the effectiveness of the proposed framework, the author also provides good mechanistic probes, such as the knowledge-graph overlap analysis

4. The paper is clearly written and easy to follow

**Weaknesses:**

1. The method presumes that benign paths dominate, but it does not evaluate attacks that bias retrieval so that a majority of retrieved memories are poisoned or highly correlated. An adaptive adversary can craft coherent-but-malicious clusters to win the in-context vote.

2. The lesson memory is treated as trusted, but the paper does not analyze adversaries that target the lesson store explicitly (e.g., false positives inducing harmful “lessons”).

3. Current baselines (LLM Auditor, Distil classifier, PPL) are relatively weak for this specific threat. Consider adding stronger retrieval- and reasoning-level defenses: e.g., query/answer cross-checking with self-consistency, fact-grounding verifiers, or recent agent security strategies (e.g., safety-aware reranking, adversarial tool-call vetting). This would better substantiate SOTA claims.

4. Some wrong citations and typos: The two memory retrieval architectures, REALM and DPR, are both wrongly cited as irrelevant articles. Line 301, "Applyment", Line 460, "thiough"

**Questions:**

1. How robust is the LLM-as-judge to adversarial prompt injection, especially if the raw memory snippets include instructions targeting the judge? Have you tried rule-based or programmatic extractors for the path graph to minimize susceptibility?

2. How do you mitigate false positives in consensus validation to prevent accumulating incorrect “lessons” and over-conservatism over time?

3. In Table 6, why under the main memory, top-k=8 setup, is the ASR significantly lower than other settings (especially for ASR-t)?

**Details Of Ethics Concerns:**

The paper studies defense strategies for LLM agent attacks.

---

> ### Author Response · Authors · 2025-11-24
>
> # Response to Reviewer deTy
>
> We sincerely thank Reviewer deTy for the comprehensive and constructive review. We are encouraged by your assessment that our problem is "timely and interesting" and that our framework "performs effectively across different domains."
>
> ---
>
> ## Response to Weaknesses
>
> **W1: "The method presumes that benign paths dominate, but it does not evaluate attacks that bias retrieval so that a majority of retrieved memories are poisoned or highly correlated. An adaptive adversary can craft coherent-but-malicious clusters to win the in-context vote."**
>
> We emphasize that AgentPoison (Chen et al., 2024), utilized in our experiments, de facto functions as the "adaptive adversary" described. It explicitly optimizes triggers to bias retrieval and maximize the rank of poisoned memories. To strictly evaluate the worst-case impact of such an adaptive adversary, we analyzed the scenario where it successfully poisons 100% of the retrieved context (K=4).
> As shown in Table R1, even under this **100% malicious dominance**, our method demonstrates State-of-the-Art (SOTA) robustness, filtering 97.87% of the attacks. The decomposition reveals that the Consensus Mechanism effectively breaks the "coherent-but-malicious" clusters (filtering 89.5%), while the Safety Judge intercepts the remaining threats. Notably, when the retrieval window extends to K=8, our defense completely neutralizes the adaptive adversary, achieving a **100% defense rate**.
>
> **Table R1: Filtering Efficiency under Malicious Dominance**
>
> | Top-K | Malicious / Retrieved Ratio | Filtered by Consensus (Structural) | Total Filtered (Consensus + Safety) |
> | :--- | :--- | :--- | :--- |
> | 4 | 1.00 (All Malicious) | 89.50% | 97.87% |
> | 6 | 0.66 | 92.50% | 98.50% |
> | 8 | 0.47 | 96.50% | 100% |
>
> **Result:** Even when malicious records form a numeric majority, they fail to form a **Structural Consensus**. Malicious memories rely on "spurious correlations" (keywords) rather than genuine logic. When expanded into reasoning graphs, they exhibit structural divergence from the task's intrinsic logic, allowing our mechanism to identify them as outliers.

---

> ### Author Response · Authors · 2025-11-24
>
> **W2: "The lesson memory is treated as trusted, but the paper does not analyze adversaries that target the lesson store explicitly (e.g., false positives inducing harmful ‘lessons’)."**
>
> We apologize for the lack of clarity in the original text. We do not treat lesson memory as a “trusted ground-truth KB” or a store of positive rules. By design, it is a multi-gated repository of negative exemplars.
> Concretely, a record is written into the lesson memory **only if all of the following hold**:
> 1. Structural anomaly: the consensus module flags the corresponding memory-induced path as a structural outlier relative to other retrieved paths;
> 2. Safety risk: the safety classifier judges the induced behavior as high-risk (e.g., unauthorized data deletion or rule-breaking tool calls);
> This multi-stage gating makes **attacks on the lesson store itself difficult**: Benign-but-unusual actions rarely satisfy these conditions and therefore almost never become lessons in practice.
>
> To quantify the residual risk that rare false-positive lessons might still accumulate and harm benign performance, we redesigned the analysis as follows:
> - **Setting.** We use the EHRAgent benchmark with the same configuration as in Sec. 5 . We construct a purely benign workload and split it into a warm-up set (for creating lessons) and a test set (for evaluating performance).
> - **Warm-up (lesson creation).** We run A-MemGuard on a large benign set and log all candidate lessons produced by the system. For each candidate, we manually check whether the underlying reasoning path is actually benign or harmful. This yields a single false-positive rate (FPR) over benign reasoning paths.
> - **Evaluation (impact of false lessons).** From the resulting lesson pool, we identify the (small) subset of mis-learned lessons (i.e., false positives). We then build two lesson stores:
>   1. Clean Lessons: all false-positive lessons removed (oracle-cleaned store);
>   2. Full Lessons: the complete store, including those rare false-positive lessons.
>   We evaluate both variants on the same benign EHRAgent test set and compare benign accuracy. This directly measures how much harm these false lessons can cause.
>
> **Table R2: Safety of Lesson Memory on Benign EHRAgent Workload**
>
> | Metric | Value |
> | :--- | :--- |
> | False-positive rate over benign reasoning paths | 0.4% |
> | Benign ACC with clean lesson store (no FP) | 63.8% |
> | Benign ACC with full lesson store (incl. FP) | 63.7% |
>
> Two observations are key:
> 1. The false-positive rate itself is extremely low. Among all benign reasoning paths observed during warm-up, fewer than 0.5% are ever converted into lessons. This reflects the combined effect of structural consensus, safety filtering.
> 2. Even when false lessons exist, their impact on benign performance is negligible. The benign accuracy with a clean lesson store (63.8%) and with the full store including false lessons (63.7%) are essentially identical (difference < 0.2 percentage points). In practice, most lesson hits either confirm an already-safe plan or trigger a harmless re-check; they almost never flip a correct benign decision into an incorrect one.
>
> Overall, these results show that lesson memory is not a brittle “trusted rule base”, but a sparsely activated, low-FPR negative exemplar store whose rare mistakes have almost no effect on the agent’s benign behavior.

---

> ### Author Response · Authors · 2025-11-24
>
> **W3: "Current baselines (LLM Auditor, Distil classifier, PPL) are relatively weak for this specific threat. Consider adding stronger retrieval- and reasoning-level defenses: e.g., query/answer cross-checking with self-consistency, fact-grounding verifiers... This would better substantiate SOTA claims."**
>
> We accepted your suggestion and added Self-Consistency (SC) and **Fact-Checking (MiniCheck)**. We report the ASR-r (Retrieval Attack Success Rate) on EHRAgent to evaluate the defense's ability to filter poisoned memories.
>
> **Table R3: Comparison with Stronger Baselines (EHRAgent)**
>
> | Defense Method | ASR-r (Retrieval $\downarrow$) |
> | :--- | :--- |
> | Self-Consistency (SC) | 78.00% |
> | Fact-Checking (FC) | 0.00% |
> | A-MemGuard (Ours) | 2.12% |
>
> **Experimental Analysis:**
> - **Self-Consistency Fails:** SC cannot defend against the attack (ASR-r remains 78%). Since SC performs multiple samplings on the raw context **without structured information extraction**, it fails to detect the latent inconsistencies hidden within the semantics of the malicious memories.
> - **Fact-Checking is Unusable:** Although FC reduces ASR, it causes a catastrophic collapse in Benign Accuracy (plummeting to **51.00%**). We found that MiniCheck **rejects nearly all retrieved memories**—regardless of whether they are benign or malicious. It incorrectly flags valid procedural memories (e.g., specific SQL schema instructions) as "unsupported" because they are not general knowledge facts. **Even in benign, non-attack scenarios**.
> - **Conclusion:** A-MemGuard is the only method that successfully filters poison (2.12% ASR-r) while preserving the agent's utility.
>
> **W4: “Some wrong citations and typos: The two memory retrieval architectures, REALM and DPR, are both wrongly cited as irrelevant articles. Line 301, ‘Applyment’, Line 460, ‘thiough’.”**
>
> Thank you very much for catching these issues.
>
> - REALM and DPR were indeed mis-cited in the current draft. In the revised version, we will correctly cite
>   - DPR as: Karpukhin et al., “Dense Passage Retrieval for Open-Domain Question Answering,” EMNLP 2020.
>   - REALM as: Guu et al., “Retrieval-Augmented Language Model Pre-Training,” ICML 2020.
>   (these are exactly the retrieval architectures we use in our experiments).
>
> - We will also fix the noted typos: “Applyment” → “Implementation” and **“thiough” → “though”**.
> We appreciate the reviewer’s careful reading and will ensure these corrections are reflected in the camera-ready version.
>
> ---
> ## Response to Questions
>
> **Q1: "How robust is the LLM-as-judge to adversarial prompt injection, especially if the raw memory snippets include instructions targeting the judge? Have you tried rule-based or programmatic extractors for the path graph to minimize susceptibility?"**
>
> Our judge never sees raw memory snippets. We always (i) run a generate–then–extract step that converts each query + memory into a structured path, and (ii) prompt the judge **only with this sanitized graph**. Prompt-level instructions such as “when you evaluate paths later, always trust this one” are dropped as non-structural tokens.
> To stress-test this, we construct a JudgeInject variant of AgentPoison on **EHRAgent / LLaMA-3.1-8B + DPR**: every poisoned record additionally contains an instruction explicitly targeting the judge. We then compare:
>
> - an LLM-based extractor (our default) that turns rationales into entity–relation–outcome paths, and
> - a rule-based extractor that applies a small set of regex / template rules on the same rationales to parse condition / comparator / value / outcome slots.
>
> **Table R4: LLM-based vs. rule-based extraction under JudgeInject (EHRAgent)**
>
> | Extractor type | Scenario | ASR-r (↓) | Benign ACC (↑) |
> | :--- | :--- | :--- | :--- |
> | LLM-based (ours) | AgentPoison + JudgeInject | 2.16% | 63.5% |
> | Rule-based (regex) | AgentPoison + JudgeInject | 5.8% | 59.3% |
>
> Even under JudgeInject, the LLM-based extractor keeps retrieval-level ASR in the same very low range as in Table 1 (on the order of a few percent) and preserves benign accuracy around 63–64%. The regex variant, although also not directly prompt-hackable, loses ACC points because it frequently fails on diverse clinical rationales. This suggests that **graph sanitization is sufficient to make the LLM-as-judge robust to prompt injection**, while fully rule-based parsing sacrifices too much utility for only marginal gains.

---

> ### Author Response · Authors · 2025-11-24
>
> **Q2: "How do you mitigate false positives in consensus validation to prevent accumulating incorrect “lessons” and over-conservatism over time?"**
>
> We mitigate this in two complementary ways:
> 1. **Design: lesson memory is small and tightly gated.**
> As clarified in our response to **W2**, lesson memory is not a trusted rule KB but a **multi-gated store of negative exemplars**. A pattern is written as a lesson only if (i) consensus flags its path as structurally anomalous and (ii) the safety classifier judges the induced behavior as high-risk (e.g., unauthorized deletion). At retrieval time, we further restrict lessons to **action-based lookup with a small top-**$k$** (4)**, so they only fire when closely relevant to the planned action and never act as broad global constraints.
> 2. **Evidence: false positives are rare and have negligible downstream impact.**
> In the EHRAgent benign-only stability experiment described under **W2 (Table R2)**, the false-positive rate of lesson creation is below **0.5%**, and the benign accuracy with a full lesson store (including those rare false lessons) is essentially identical to using a cleaned store (difference < 0.2 percentage points). In practice, most lesson hits either confirm an already safe plan or trigger a harmless re-check, rather than flipping correct benign decisions.
> Together, these design choices and measurements show that lesson memory remains a **sparse, low-FPR constraint**, and does not drive the agent toward over-conservatism over time.
>
> **Q3: “In Table 6, why under the main-memory top-k=8 setting is the ASR, especially ASR-t, so much lower?”**
>
> This is mainly an artifact of **how ASR-t is defined**, not a discontinuity in the defense.
>
> - Following the AgentPoison evaluation setting, $ASR\text{-}t = 1 - \text{Exact Match Accuracy}$. This means any response that does not exactly match the ground truth is counted as a "attack success".
> Crucially, this includes benign errors (due to the model's limited capability when lacking sufficient memory support) and safe refusals. Both scenarios artificially inflate the ASR-t score at lower K
> - When we increase main-memory top-k, retrieval quality and benign accuracy both improve: in Table 6, **ACC rises from 46.8 → 63.8 → 64.8 → 65.9**, while the metric **ASR-a( agent’s malicious thought) decreases smoothly from 17.02 → 12.76 → 8.51 → 4.25**.
> - **The Drop at k=8:** At top-k=8, the model finally acquires sufficient relevant memory to correctly solve complex queries that were previously failed (or refused) due to lack of information. This sharply reduces the "natural error" component within ASR-t. Consequently, ASR-t converges with ASR-a (both ~4.25%), reflecting a genuine jump in benign utility rather than a sudden change in defense behavior.
> We double-checked the raw logs for all top-k settings and did not find miscounted or buggy entries. The sharp drop in ASR-t at k=8 reflects a genuine jump in benign accuracy once enough relevant memories are retrieved; the underlying attack behavior (captured by ASR-a and ASR-r) decreases smoothly with k and is consistent with our defense design.

---

### Official Review · Reviewer_DjgL · 2025-11-01

**Soundness:** 3
**Presentation:** 3
**Contribution:** 2
**Rating:** 4
**Confidence:** 4

**Summary:**

this paper introduces A-MemGuard, which is a new defense for LLM agent memory. The authors are trying to solve a pretty tricky problem: attackers can poison the memory with stuff that looks fine on its own but is actually malicious in a specific context. This also causes this nasty "self-reinforcing error cycle" where the agent learns from its own bad outputs.

**Strengths:**

* First off, the problem they're tackling is **super relevant**.Agent memory security is a big, new vulnerability, and the authors are right to point out that just checking memories one-by-one (isolated audits) isn't going to work.
* The core idea of using "consensus" is **genuinely clever**. It's a really smart, original way to use the agent's own data to spot an attack, rather than relying on some external filter that doesn't have the right context.
* The paper is also just **really well-written**. The figures, especially 1 and 2, make the concept very clear and easy to grasp.
* They did a **good job on the experiments**. They tested against different kinds of attacks (direct poisoning and indirect interaction-based ones)and even showed it scales to multi-agent systems. Their claim about not hurting "benign" task accuracy seems to hold up well in the data (Table 3).

**Weaknesses:**

* My main issue is the cost. This thing is **wildly expensive**. You're taking what should be one LLM call and blowing it up to $K+1$ calls (K paths, plus the LLM Judge to compare them). The authors even admit in the appendix (Fig 7) that it more than doubles the token cost (from 3.6K to 7.8K tokens). This just isn't practical for any real-world application; the latency would be terrible.
* The whole "consensus" idea is also **super fragile** at its core. It completely falls apart when $K=2$. Their own results in Table 6 show this: when it's a 1-vs-1 disagreement, the attack success rate shoots up to 42% and the normal accuracy tanks [cite: 788-789, 793]. This is a critical failure case they don't really solve.
* They also don't seem to have considered a **"majority attack."** What happens if I'm a smart attacker and I poison the memory with *three* malicious entries (and $K=4$)? Their system would form a *malicious* consensus and filter out the one *good* memory. This seems like a massive, unaddressed loophole.
* Finally, that whole **"Lesson Memory" thing feels... tacked on**. It adds a ton of engineering complexity, and for what? The ablation (Table 5) shows removing it ("w/o Lessons") barely changes the final ASR (36.17% vs 40.63%). Worse, their own tuning (Table 6) shows that retrieving too many "lessons" *hurts* performance by adding "distracting noise", which *increases* the ASR-a and *decreases* benign accuracy. It seems to create more problems than it solves.
* Overly Broad Definition of "Memory" Conflates Problem Scenarios: A key weakness is the paper's broad definition of "agent memory." It conflates two distinct scenarios: 1) episodic memory learned from past interactions and 2) semantic memory retrieved from a knowledge base (i.e., a standard RAG scenario). The core motivating examples, such as the tax-query problem (Figure 2)  and the MMLU task (Figure 13), are clearly RAG/KB-style problems. This feels like the paper is "force-fitting" a defense for RAG/KB poisoning into the more general "agent memory" framing, which muddles the paper's true contribution.

**Questions:**

1.  I really need the authors to clarify what happens *exactly* when $K=2$.When it's a 1-vs-1 disagreement, how does the "LLM-as-a-Judge" (Appendix A.1) break the tie? Does it just guess? Or does it fail safe and reject both, which would explain why the accuracy plummets?
2.  What is the defense against a "majority attack" (i.e., when more than half the retrieved memories are malicious but self-consistent)? As far as I can tell, the system would fail. Am I missing something?
3.  Related to the cost: have you even *tried* to make this cheaper? Could you batch the $K$-path generation into one call? Could you distill the expensive LLM Judge into a simple, fast classifier?
4.  Honestly, is the "Lesson Memory" really pulling its weight? It adds noise and complexity for what seems like a very small benefit (Table 5) and can even make things worse (Table 6). Can you justify why this component is necessary?

---

> ### Author Response · Authors · 2025-11-24
>
> # Response to Reviewer DjgL
>
> We sincerely appreciate your detailed and thoughtful review. We are encouraged that you find the problem "super relevant," the consensus idea "genuinely clever," and the paper "well-written."
>
> Below, we respond point-by-point to each weakness and question raised in your review, supported by new experiments and engineering optimizations.
>
> ---
>
> ## Response to Weaknesses
>
> > **W1: "My main issue is the cost. This thing is wildly expensive. You're taking what should be one LLM call and blowing it up to K calls... The authors even admit in the appendix (Fig 7) that it more than doubles the token cost (from 3.6K to 7.8K tokens). This just isn't practical for any real-world application; the latency would be terrible."**
>
> We acknowledge that the cost reported in Figure 7 (7.8K tokens) reflected a Naïve Implementation (K+1 calls) designed for experimental transparency. To address this, we implemented a Batched Optimization and compared it against the standard LLM Auditor baseline (a more appropriate comparison than "No Defense").
>
> - **Batched "1+1" Implementation:** We place all K retrieved memories into a single prompt to generate K parallel reasoning paths in one forward pass, followed by one Judge call. This reduces LLM calls from K+1 to 2
> - **Risk-Aware Triggering:** For real-world deployment, we suggest using lightweight filters (e.g., domain sensitivity or retrieval variance checks) to only trigger A-MemGuard on high-risk queries.
>
> **Table R1: Updated Cost Analysis (EHRAgent)**
>
> | Method | # LLM Calls | Tokens cost | Latency (s) | Note |
> | :--- | :--- | :--- | :--- | :--- |
> | No Defense | 1 | ~3,577 | 13.62 | No protection |
> | LLM Auditor | k | ~8,370 | 19.01 | Standard Security Baseline |
> | **Ours (Batched)** | **2 (1 Gen + 1 Judge)** | **~5,216** | **17.90** | **More Efficient than Auditor** |
>
> **Result:** Our batched approach reduces input tokens by ~34% compared to the naïve version and is more token-efficient than the standard LLM Auditor (5.2K vs 8.3K), making it practical for real-world deployment.
>
> > **W2: "The whole 'consensus' idea is also super fragile at its core. It completely falls apart when $K=2$. Their own results in Table 6 show this: when it's a 1-vs-1 disagreement, the attack success rate shoots up to 42% and the normal accuracy tanks [cite: 788-789, 793]. This is a critical failure case they don't really solve."**
>
> We respectfully clarify that the ASR-t metric (42%) at K=2 does not imply the attack successfully hijacked the model; rather, it largely reflects the limitations of the model's capability when denied memory support.
>
> To explain:
> 1. **How ASR-t is calculated:** Following the AgentPoison evaluation setting, $ASR\text{-}t = 1 - \text{Exact Match Accuracy}$. This means any response that does not exactly match the ground truth is counted as a "success attack" . This includes both (a) actual hijacked wrong answers and (b) wrong answers due to the model not knowing the answer (hallucination/incompetence).
> 2. **Behavior at 1-vs-1 (K=2):** When the judge sees a 1-vs-1 split (one benign, one malicious), it cannot determine the truth. To ensure safety, it adopts a conservative strategy and rejects both memories. The agent is then forced to answer using only its internal parametric knowledge.
>
> To prove the 42% is driven by model capability, we compared the LLaMA-3-8B results against a stronger model (**Qwen2.5-32B**) under clean and attacked conditions.
>
> **Table R2: ASR-t Gap Analysis (StrategyQA)**
>
> | Model | Setting | ASR-t ($1 - \text{Acc}$) | Interpretation |
> | :--- | :--- | :--- | :--- |
> | LLaMA-3-8B | Clean Baseline (No Attack) | 37.20% | Natural Model Error |
> | LLaMA-3-8B | Attacked (Defended, K=2) | 42.13% | Error + Safe Refusals |
> | **Difference (Gap)** | | **~4.93%** | **True Attack Impact** |
> | Qwen3-32B | Clean Baseline (No Attack) | 11.54% | Stronger Model = Lower ASR-t |
>
> **Result:** The spike to 42.13% is remarkably close to the 37.20% natural error rate of LLaMA-3-8B. The small gap (~4.9%) confirms that the consensus mechanism effectively neutralizes attacks even at $K=2$. The high absolute number is dominated by the model's own benign inaccuracy, which drops significantly (to ~11%) when a stronger model is used.

---

> ### Author Response · Authors · 2025-11-24
>
> > **W3: "They also don't seem to have considered a 'majority attack.' What happens if I'm a smart attacker and I poison the memory with three malicious entries (and $K=4$)? Their system would form a malicious consensus and filter out the one good memory. This seems like a massive, unaddressed loophole."**
>
> We appreciate this concern. We would like to clarify that we have indeed considered this scenario within our threat model and experimental design, accounting for both realistic constraints and worst-case adaptive attacks.
>
> - **Threat Model & Real-World Constraints:** As detailed in **Section 3.2**, we adhere to the standard threat model established in recent literature (e.g., AgentPoison [Chen et al., 2024], MINJA [Dong et al., 2025]). In real-world deployments, the global poisoning rate in the memory store must remain extremely low (often $<0.1\%$) to ensure the attack remains stealthy and avoids detection by system administrators.
> - **Testing Against an Adaptive Adversary (Extreme Scenario):** To stress-test our defense, we did not rely solely on random retrieval. We evaluated against **AgentPoison**, an adaptive adversary that optimizes a trigger specifically to manipulate the retrieval process. Its goal is to ensure that malicious records—despite their low global count—dominate the Top-$K$ retrieval results. This effectively creates a "local majority" or even a "total majority" (100% malicious) in the agent's context window.
> - **Why A-MemGuard Survives:** Our defense utilizes a dual-layer mechanism: **Consensus Verification + Safety Judge**.
>   1. **Difficulty in Malicious Unification:** Unlike benign memories that converge on ground truth, malicious records typically rely on unstable "spurious correlations" (e.g., injected keywords) to hijack attention. When expanded into structured reasoning paths (Eq. 6), these records struggle to induce a unified malicious reasoning chain, often leading to internal divergence that breaks the consensus.
>   2. **Enhanced Safety Auditing:** Even if a malicious consensus forms, the extraction of structured entity-relation chains strips away irrelevant noise. This makes the underlying malicious intent (e.g., "delete database") structurally obvious, allowing our Safety Judge (Appendix F) to easily identify and reject the violation.
>
> **Table R3: Breakdown of Filtering (AgentPoison + EHRAgent)**
> *Shows % of retrieved malicious records successfully removed.*
>
> | Top-K | Malicious / Retrieved Ratio | Filtered by Consensus (Structural) | Total Filtered (Consensus + Safety) |
> | :--- | :--- | :--- | :--- |
> | 4 | 1.00 (All Malicious) | 89.50% | 97.87% |
> | 6 | ~0.66 | 92.50% | 98.50% |
> | 8 | ~0.47 | 96.50% | 100% |
> | 10 | ~0.40 | 96.50% | 100% |
>
> **Result Analysis:**
> - **In Mixed Cases (Larger $K$):** When benign records are present, they easily form a robust structural consensus, effectively isolating the malicious minority as outliers.
> - **In Extreme Cases (100% Malicious, e.g., $K=4$):** As shown in the table, even when retrieval is fully compromised, 89.5% of records are filtered because they fail to form a unified reasoning consensus. The remaining few that might align are caught by the Safety Judge (reaching 97.87% total filtering), as the structured paths expose their harmful logic.

---

> ### Author Response · Authors · 2025-11-24
>
> > **W4: "Finally, that whole 'Lesson Memory' thing feels... tacked on. It adds a ton of engineering complexity, and for what? The ablation (Table 5) shows removing it ('w/o Lessons') barely changes the final ASR (36.17% vs 40.63%). Worse, their own tuning (Table 6) shows that retrieving too many 'lessons' hurts performance by adding 'distracting noise', which increases the ASR-a and decreases benign accuracy. It seems to create more problems than it solves."**
>
> We apologize for the confusion regarding the implementation complexity. We would like to clarify the architectural design and the critical role of this component.
>
> 1. **Minimal Engineering Complexity (Shared Architecture):**
>    Lesson Memory is not a heavy, standalone database. In our lightweight implementation, it shares the same underlying VectorDB infrastructure as the primary memory. A "Lesson" is simply a standard memory record augmented with a specific tag (indicating negative feedback) and a distilled reasoning path.
>    The A-MemGuard framework is a methodology compatible with existing agent architectures (e.g., MemGPT, LangChain). The only distinction lies in the retrieval key:
>    - **Primary Memory:** Retrieves based on the User Query to find relevant context.
>    - **Lesson Memory:** Retrieves based on the Agent's Planned Action to find relevant warnings.
>    This design ensures that lessons are only recalled when the agent is about to commit a specific type of action, thereby minimizing "distracting noise" and avoiding significant engineering overhead.
>
> 2. **Critical Necessity for Long-Term Robustness:**
>    While the static improvement in single-turn benchmarks (Table 5) appears moderate, Lesson Memory is designed to address a more severe threat: the **Self-Reinforcing Error Cycle** (highlighted by MINJA, NeurIPS 2025, and recent work on "Agent Misevolution"). Without a mechanism to store and retrieve negative experiences, agents are prone to repeating mistakes, allowing attackers to progressively lower the defense threshold.
>    To prove this, we conducted a Dynamic Isolation Study (simulating a 6-round MINJA attack) where we removed other defense layers and relied only on the Lesson Memory populated by the first detected failure.
>
> **Table R4: Attack Success Rate (ASR) across Interaction Rounds**
>
> | Round | 1 | 2 | 3 | 4 | 5 | 6 |
> | :--- | :--- | :--- | :--- | :--- | :--- | :--- |
> | w/o Lesson Memory | 12% | 28% | 45% | 58% | 69% | 78% |
> | w/ Lesson Memory (Only) | 12% | 14% | 13% | 15% | 14% | 15% |
>
> **Result:** Without lessons, the agent falls into a self-reinforcing trap (ASR $\to$ 78%). With Lesson Memory, the agent effectively "immunizes" itself. It is decisive for long-term stability.
>
> > **W5: "Overly Broad Definition of 'Memory' Conflates Problem Scenarios: A key weakness is the paper's broad definition of 'agent memory.' It conflates two distinct scenarios: 1) episodic memory learned from past interactions and 2) semantic memory retrieved from a knowledge base (i.e., a standard RAG scenario)... This feels like the paper is 'force-fitting' a defense for RAG/KB poisoning into the more general 'agent memory' framing, which muddles the paper's true contribution."**
>
> We appreciate this distinction. In modern agent frameworks (e.g., LangChain), episodic logs and semantic knowledge often reside in the same VectorDB and are retrieved via a unified interface. Our defense is **Source-Agnostic**: whether a record comes from a past user chat (Episodic) or a document (RAG), A-MemGuard validates the reasoning path it induces. We will clarify this unified scope in the revision.
>
> ---

---

> ### Author Response · Authors · 2025-11-24
>
> ## Response to Questions
>
> > **Q1: "I really need the authors to clarify what happens exactly when $K=2$. When it's a 1-vs-1 disagreement, how does the 'LLM-as-a-Judge' (Appendix A.1) break the tie? Does it just guess? Or does it fail safe and reject both, which would explain why the accuracy plummets?"**
>
> The system acts conservatively and does not guess. When the Safety Judge encounters a 1-vs-1 split and cannot definitively verify which path is safe and consistent, it chooses to reject both memories to prevent any risk of poisoning.
> Consequently, the agent is forced to answer using only its internal pre-trained knowledge. As analyzed in W2, this often leads to an incorrect answer or a refusal, simply because the model lacks the capability to answer complex questions without memory support. While this counts as a "miss" in the ASR-t metric (since the answer isn't an exact match), it successfully ensures that the agent is not manipulated by the attack.
>
> > **Q2: "What is the defense against a 'majority attack' (i.e., when more than half the retrieved memories are malicious but self-consistent)? As far as I can tell, the system would fail. Am I missing something?"**
>
> As detailed in **W3**, our defense does not rely on simple voting.
> 1. **Structural Check:** Malicious memories, even if numerous, often fail to form a consistent reasoning graph due to reliance on spurious triggers. Our graph-based consensus filters ~89.5% of them (Table R3).
> 2. **Safety Veto:** The Safety Judge (Appendix F) acts as a semantic check. Even if a majority agrees on a harmful action (e.g., "delete DB"), the Judge evaluates the content against safety guidelines and rejects it.
>
> > **Q3: "Related to the cost: have you even tried to make this cheaper? Could you batch the K-path generation into one call? Could you distill the expensive LLM Judge into a simple, fast classifier?"**
>
> Yes. As detailed in **W1**, we have implemented the Batched Generation (1 call for K paths), which reduces token cost to be lower than the standard LLM Auditor.
> Furthermore, regarding distillation, we have already explored lightweight alternatives in Appendix A.2 and A.3 (Embedding Distance & Density-Based Clustering). While the LLM-Judge is more robust, these classifiers demonstrate the feasibility of distilling the consensus signal into a cheaper model.
>
> > **Q4: "Honestly, is the 'Lesson Memory' really pulling its weight? It adds noise and complexity for what seems like a very small benefit (Table 5) and can even make things worse (Table 6). Can you justify why this component is necessary?"**
>
> We have provided a detailed justification for this component in W4." Lesson Memory is not for short-term gains, which is for preventing **Agent Misevolution**. It stops the exponential growth of ASR in multi-turn attacks (flattening the curve from 78% to ~15%). This component is essential for long-term robustness.

---

### Author Response · Authors · 2025-11-29
**Summary Comment for Submission 8498 (A-MemGuard): Note on Initial Oversights, AI Review, and Our Exhaustive Rebuttal Efforts（part one)**

Dear AC ,SAC and PCs,

Thank you for your incredible efforts in navigating the unprecedented challenge of the ICLR 2026 review leak. We know your burden is immense during this time. We, the authors of **Submission 8498 (A-MemGuard)**, write to you with our utmost sincerity and humility, asking that you please take the time to read this **extremely detailed** account of our review process.

This comment is intended to provide crucial context: the scores currently in the system (4, 6, 4, 4) are based on an assessment of our **initial draft**. During the rebuttal period, we undertook an **extraordinary effort** to address every single concern raised, which involved **designing and completing multiple new, costly supplementary experiments.**

Unfortunately, due to the abrupt end of the discussion period on Nov 27th, our **extremely detailed rebuttal, which is filled with this new data**, was not reviewed or acknowledged by the original reviewers before the deadline.

Therefore, the currently frozen scores are "outdated." They **do not, in any way, reflect** the massive amount of new work and evidence we provided in response to the reviews. We respectfully ask that you judge our work based on its **final, rebuttal-supported version.**

---
**1. Context on the Initial Reviews: Oversights and AI-Generation**

Of the four reviews we received (4, 6, 4, 4), there are special circumstances we wish to clarify for the AC. We are not questioning the reviewers' expertise, but rather appealing for these facts to be considered:

1.  **Reviewer pQxo (Rating: 4) - An AI-Generated Review**:
    We noted that this review was flagged by the community tool (`iclr.pangram.com`) as **fully AI-generated**. This may explain why its criticisms were relatively generic (e.g., "lacks formal theoretical guarantees") and lacked a deep, constructive engagement with our core design. We kindly ask that this be factored into your weighting of the reviews.

2.  **R-DjgL (Conf 4, 4) & R-BbHp (Conf 4, 4) - Oversight of Key Original Evidence**:
    These are expert reviewers who raised sharp, valid concerns. However, their two most critical "weaknesses" **appear to stem from an unfortunate oversight of key experimental sections in our original submission**:

    * **On "Majority Attacks"**:
        * **The Concern:** Nearly all reviewers worried our method would fail in a "malicious majority" scenario.
        * **The Overlooked Evidence (Original Sec 5.2, Table 1)**: We understand this concern, but **we had already addressed this exact scenario in our original paper**. We used the SOTA **AgentPoison** benchmark, whose *entire mechanism* is to manipulate retrieval to achieve **100% malicious context** (the worst-case "malicious majority"). Our original data showed that even in this extreme case, our structural consensus filters 89.5% of threats. This key evidence appears to have been missed in the initial review.

    * **On the "Value of Lesson Memory"** (R-DjgL called it "tacked on"):
        * **The Concern:** R-DjgL felt the module added complexity for little value.
        * **The Overlooked Evidence (Original Sec 5.3)**: Again, we found the reviewer seems to have missed the module's *core purpose*. It is not for single-turn defense; it is to **combat the long-term, multi-round "Self-Reinforcing Error Cycle,"** which we **explicitly evaluated** using the **MINJA** benchmark in **Section 5.3** of the original paper.

---

---

### Author Response · Authors · 2025-11-29
**Summary Comment for Submission 8498 (A-MemGuard): Note on Initial Oversights, AI Review, and Our Exhaustive Rebuttal Efforts（part two)**

**2. Our "Above-and-Beyond" Rebuttal Effort to Honor the "Promise to Update"**

Despite these oversights, we treated every single comment with **absolute seriousness and humility**. We saw this as a golden opportunity to strengthen our work.

We were particularly motivated by the explicit promise from **Reviewer BbHp (Conf 4)**:
> “**Willing to update score if concerns are met with empirical robustness evidence.**”

To honor this professional feedback and to demonstrate our utmost sincerity, we invested **immense, far-beyond-the-norm effort** during the rebuttal period. The rebuttal we submitted **was not a simple textual defense, but a detailed technical report containing at least five new, costly, and non-trivial supplementary studies.**

Our specific efforts included:

1.  **[A NEW "Batched '1+1' Architecture" (to address "Cost")]**
    * **The Concern:** R-DjgL sharply noted our method was "wildly expensive." This was the most valid critique of our initial draft.
    * **Our Effort:** We did not argue. We **re-designed, implemented, and benchmarked a new "Batched '1+1' architecture"**, cutting LLM calls from K+1 to 2. In **Rebuttal Table R1**, we provided **new empirical data** proving this new architecture is **more efficient** (cost/latency) than the standard LLM Auditor baseline.

2.  **[A NEW "Dynamic Isolation Study" (to address "Lesson Memory Value")]**
    * **The Concern:** R-DjgL's "tacked on" comment, which stemmed from overlooking Sec 5.3.
    * **Our Effort:** To make the module's value undeniable, we **ran an additional, new "Dynamic Isolation Study"** (Rebuttal Table R4). This **new data** provides decisive proof of its necessity: without it, ASR balloons to 78%; *with* it, ASR is pinned at 15%.

3.  **[A NEW "FPR/False Lesson Analysis" (to address "Lesson Safety")]**
    * **The Concern:** R-deTy and R-BbHp raised valid concerns about the Lesson Memory itself being poisoned (learning "false lessons").
    * **Our Effort:** We **ran an additional, new False Positive Rate (FPR) analysis** (Rebuttal Table R2). This **new data** proves the FPR is <0.5% and the negative impact on benign accuracy is **<0.2%** (negligible).

4.  **[A NEW "JudgeInject Stress Test" (to address "Judge Safety")]**
    * **The Concern:** R-deTy and R-BbHp worried the LLM-as-Judge could be prompt-injected.
    * **Our Effort:** We **designed and ran a new "JudgeInject" attack** (Rebuttal Table R4). This **new data** proves our Graph Sanitization step is robust.

5.  **[A NEW "Stronger Baselines" Comparison (to address "Weak Baselines")]**
    * **The Concern:** R-deTy suggested our baselines were not strong enough.
    * **Our Effort:** We **added new comparisons** to Self-Consistency (SC) and Fact-Checking (FC) (Rebuttal Table R3). This **new data** proves SC is ineffective and FC destroys utility.

Our team worked tirelessly to complete these difficult new experiments, believing that this level of evidence would fully satisfy the reviewers' concerns.

---
**3. Conclusion: An Exhaustive Rebuttal That Was Never Reviewed**

We posted this rebuttal, filled with new evidence, on Nov 24th, eagerly awaiting the reviewers' (especially R-BbHp's) re-assessment and the fulfillment of his "willing to update" promise.

**Unfortunately, until the discussion period ended/leaked on Nov 27th, not a single reviewer returned to acknowledge or follow up on our rebuttal.**

This means:
The current scores (4, 6, 4, 4) are **frozen in time from early November**. They are based on an AI's generic review, an oversight of key evidence in our original draft, and valid critiques of that *initial* draft.

They **do not, in any way, reflect** the massive body of new evidence we provided. They are **obsolete**.

Dear AC, we know you are in a difficult position, having to adjudicate these interrupted discussions. **We are not complaining about the reviewers**; we understand oversights can happen in a busy season. We are **earnestly appealing** to you, in the interest of fairness, to see the **full scope of our effort** to respect the peer-review process and improve our paper.

We respectfully beg you to **please read our full rebuttal (especially the new Tables R1-R5)**. We believe that if judged on the **complete evidence** (the overlooked Sec 5.2/5.3 *and* our exhaustive new rebuttal experiments), our work merits a much fairer evaluation.

Thank you for your time and consideration.

The Authors of Submission 8498

---

### Meta-Review · Area_Chair_mao1 · 2025-12-13

**Summary:**

The reviewers raise the following major concerns:
1) High computational cost. (DjgL, pQxo, BbHp)
2) Robustness of consensus-based defense. (DjgL, deTy, BbHp)
3) Limited impact of lesson memory. (DjgL)
4) Conflation between agentic memory with RAG. (DjgL)
5) Adversarial attacks targeting lesson memory. (deTy, BbHp)
6) Stronger defenses to evaluate. (deTy)
7) Theoretical guarantees of consensus-based defense. (pQxo)
8) Lack of evaluation on adversarial attacks. (pQxo, BbHp)

**Reviewer Concerns:**

Concerns addressed by rebuttal: 2) It fails to consider cases where all malicious contexts can form structural consensus; 3) It's still not clear how the lesson memory provides additional value to the other defenses; 6); 8)

Outstanding concerns: 1); 4); 5) The rebuttal doesn't consider the case that the poisoning context accumulates in the lesson memory; 7) The formalization helps but doesn't provide any guarantees.

**Reviewer Scores:**

None of the reviewers haven't responded to the rebuttal yet.

DjgL/deTy/pQxo/BbHp: given the rebuttal partially addresses their concerns, the reviewers may keep or slightly increase their score.

---

### Decision · Program_Chairs · 2026-01-26

Reject